# Role of Fusion Imaging in Image-Guided Thermal Ablations

**DOI:** 10.3390/diagnostics11030549

**Published:** 2021-03-19

**Authors:** Serena Carriero, Gianmarco Della Pepa, Lorenzo Monfardini, Renato Vitale, Duccio Rossi, Andrea Masperi, Giovanni Mauri

**Affiliations:** 1Postgraduate School of Radiodiagnostics, Università degli Studi di Milano, 20122 Milan, Italy; gianmarco.dellapepa@unimi.it (G.D.P.); renato.vitale@unimi.it (R.V.); duccio.rossi@unimi.it (D.R.); 2Radiology Department, Fondazione Poliambulanza Istituto Ospedaliero, 25124 Brescia, Italy; lorenzo.monfardini@poliambulanza.it; 3Division of Interventional Radiology, IEO, European Institute of Oncology IRCCS, 20141 Milan, Italy; andrea.masperi@ieo.it (A.M.); vanni.mauri@gmail.com (G.M.); 4Department of Oncology and Hematology-Oncology, University of Milan, 20122 Milan, Italy

**Keywords:** fusion imaging, imaging-guided ablation, thermal ablation, image processing, ultrasound, contrast-enhanced ultrasound (CEUS), computed tomography, magnetic resonance imaging, liver, kidney, lung, musculoskeletal, prostate

## Abstract

Thermal ablation (TA) procedures are effective treatments for several kinds of cancers. In the recent years, several medical imaging advancements have improved the use of image-guided TA. Imaging technique plays a pivotal role in improving the ablation success, maximizing pre-procedure planning efficacy, intraprocedural targeting, post-procedure monitoring and assessing the achieved result. Fusion imaging (FI) techniques allow for information integration of different imaging modalities, improving all the ablation procedure steps. FI concedes exploitation of all imaging modalities’ strengths concurrently, eliminating or minimizing every single modality’s weaknesses. Our work aims to give an overview of FI, explain and analyze FI technical aspects and its clinical applications in ablation therapy and interventional oncology.

## 1. Introduction

With recent advances in technology, fusion imaging (FI) techniques in image-guided ablations have become more and more applied in clinical practice [1,2,3].

Image-guided thermal ablation (TA) is an effective treatment for several kinds of cancers. TA is considered a curative treatment for hepatocellular carcinoma (HCC) comparable to resection, for liver metastases unresponsive to chemotherapy, for local control of abdominal malignancies (non-functioning adrenal tumors, abdominal sarcomas, renal carcinomas) and extra-abdominal malignancies (bone tumors and lung primary lesions or metastases).

For the ablation success, an essential prerequisite is an optimal imaging technique to maximize the efficacy of pre-procedure planning, intraprocedural targeting, post-procedure monitoring and assessing the result achieved.

Ultrasound (US) is the most diffused technique for guiding percutaneous procedures since it does not require ionizing radiation and is a real-time, low-cost, multiplane and largely accessible technique. However, US has a low contrast resolution and a relevant number of malignancies cannot be completely detectable on US, like hepatic or renal lesions [4,5]. A further US limitation is for lesions located in the hepatic dome or behind lung shadow through an intercostal approach which could not be easily identified for a safe and effective needle placement. Contrast-enhanced Computed Tomography (CECT) and magnetic resonance imaging (MRI) have a high-contrast resolution. However, these modalities require specific high-cost equipment, often need the use of intravenous contrast, are burdened by a limited or absent real-time capability and have other specific limitations, like the scarce availability of MRI equipment for intervention and the use of ionizing radiations for CECT.

Contrast-enhanced US (CEUS) can increase the visibility of small lesion, is a cost-effective technique, but require specific operator experience and still is not widely used [6]. Moreover, CEUS still could not overcome some of the limitations described, such as the difficult lesion locations.

Developments in computer graphics and technology have permitted improvements in 3D image-processing techniques and spatial resolution, allowing one to overcome the technical boundaries of FI and making possible its adoption in everyday clinical practice.

The purpose of FI in ablative treatment is to improve target lesion identification, needle guidance and visualization of the enclosing structure. FI systems allow one to merge the information deriving from different imaging modalities, improving all steps of ablative procedures and are generally integrated in all US systems, at a relatively limited cost.

Our review aims to analyze and explain the FI technique and its application in ablation therapy and interventional oncology.

## 2. Literature Research

A systematic review was conducted, two authors (S.C. and G.D.P.) executed a literature search independently using EMBASE to identify the most relevant article on FI and its applications in interventional oncology. Search terms used were “Fusion Imaging”, “Ultrasound”, “Nuclear Magnetic Resonance”, “Computed assisted Tomography”, “Virtual Navigation” and “Interventional Radiology”. Abstracts were screened and full-text articles were obtained. All articles were independently analyzed and evaluated by both reviewers for eligibility. No publishing date limitations were imposed. Attention was placed to reduce the overlapping populations in case of multiple publications from the same center. The total number of articles screened was 110 and 62 were selected.

## 3. Fusion Imaging Techniques

To understand the role of FI and future applications is essential to know fusion technique.

There are two possible image registration and fusion approaches: the “software” and “hardware” approach. In the software approach, images are acquired on separate devices, imported into image-processing workstation and registered and fused using dedicated software.

In the hardware approach, different images are acquired on a single device, which can provide multi-modality acquisition and registered and fused with the integrated software [7].

The FI process is typically a two-stage in which the different steps are first registration and then fusion of the previously registered images [8].

### 3.1. Registration

The first step is reformatting the primary and the secondary images in the same format. The second step is to transform the secondary images set to spatially align it in three dimensions with the primary image set. The transformations can be “rigid” or “non-rigid”. In rigid transformation, the distance between two points is always constant, allowing only translation and rotation of the image set. In non-rigid transformations, selected sub-volumes within the image set may be expanded or contracted, translations and rotations may be performed as well [9]. Non-rigid registration warps and deforms the original data; it may introduce wrong assumptions, so the operator has to be careful to verify the accuracy when using this method [10].

Rigid transformations may be either manual or automated, otherwise non-rigid transformations cannot be implemented manually and are always automated.

### 3.2. Fusion

Image fusion, or co-display, is the integrated display of registered images. It can be achieved with the simultaneous display of images side-by-side or overlaying the registered images displayed in different color tables. Registration and fusion can be automated, semiautomated or manual, with different grades of user interaction. The automatic ones can be organ-dependent or modality-dependent because there are no universal algorithms of fusion. The final step in the FI process is to evaluate the accuracy of registration that can be performed by examining the position of fiducial patches or anatomical landmarks [10].

#### 3.2.1. Manual Registration Procedure

There are different modalities to perform the manual registration procedure. The most common registration procedure is the “plane and point” registration. Plane registration is the process of finding the same plane in both image data sets. To perform plane registration, any plane that clearly shows the same anatomic landmarks on both data sets can be used. Point registration, after finding the right plane, is usually performed by pointing out the same anatomic point in the two imaging modalities to be coregistered [11,12]. This procedure matches the two image data sets more accurately by reducing the spatial error in the three spatial coordinates. The whole procedure can be repeated until optimal image fusion is obtained [13].

There are other modalities to carry out the manual registration procedure, depending on different vendors software. The “rotate” or “drag”, for example, is used to integrate US images with Computed Tomography (CT) or MRI images and allows the operator to align the two image sets manually. Another manual image fusion modality is performed by matching three internal markers on any image plane [13].

#### 3.2.2. Automatic Registration Procedure

Since there are no universal algorithms, the automatic procedures are organ-dependent and modality-dependent [10]. The auto-registration algorithm works simultaneously on both image sets volume and it is based on extracting and matching regions of interest in both modalities (usually anatomical structures like vessels). It should be precise enough and at the same time fast, so the automatic registration step can be repeated during the examination if occur something that modifies the previous baseline condition during the procedure (i.e., the patient moves) [11].

Some vendors provide locating device that can perform an automatic active tracking of the patient like the CIVCO omniTRAX^™^ (Civco Medical Instruments Co Inc. Kalona, IA, USA). This tool provides automatic image registration of fused images when using real-time US with previously acquired CT volume data sets and consists of a disposable locating device and an electromagnetic sensor. An example of a system for FI in a liver image guided ablation is shown in Figure 1.

## 4. Clinical Applications

### 4.1. Liver

Image-guided tumor ablation is a consolidated, locoregional treatment for HCC that provides good therapeutic outcomes, optimal local tumor control and overall survival [14]. Furthermore, TA is mini-invasive, effective and relatively low cost [15]. FI techniques are applied in tumor ablation because they improve hepatic lesions detection: small HCC have low sonographic contrast and conspicuity than larger HCC; this is a limit in US-guided ablation alone [16]. The detectability of small HCCs at US is based on tumor size and localization in the liver [17]. FI overcomes US-limitations displaying real-time synchronized TC/MRI images in the same plane of US, obtaining an increase in detection of small cancer (<2 cm) [18]. Moreover, FI is useful in detecting atypical HCC because the combination of US and hepatobiliary phase Gd-EOB-DTPA enhanced MRI is more sensitive than conventional CEUS [18,19]. Calandri et al. after an extensive review on such topic gave their personal view on the collected data: to overcome small HCC reduced visibility, CEUS should be preferred in case of hypervascular behavior as first choice guidance technique and FI left for hypovascular lesions or in case of obstructed lesion visualization [13].

MRI and CT are both used as reference data set for FI; FI imaging with conventional US and liver CT/MR images improves the detection rate of HCC of 45% and, therefore, the procedure’s feasibility after FI [14,20]. However, some studies have shown the superiority of MRI with hepatobiliary phase to CT images for the identifications and definitions of hepatic lesions, especially in small lesions less than 1 cm [21]. Nevertheless, very promising preliminary data are available regarding US/Cone beam CT (CBCT) FI [22]. It is technically feasible and appears to be an effective image guidance modality for achieving correct targeting and ablation of small lesions not clearly visible at US (Figure 2).

This method bears the potential to overcome the major limitations of US/CT or US/MRI FI since CBCT is performed in the Angio room with the patient in the desired position for ablation despite of TC or RM images pre-acquired days before the procedure [22].

Imaging guided TA has an increasingly important role in the management of patients with liver metastases, FI increases the visibility of hepatic lesions than conventional US [23]. A study of 58 liver metastases demonstrated that percutaneous ablation of 18FDG-PET/CT positive liver lesion using FI of real-time US and previous PET/TC images facilitates the procedures that could be impossible and challenging with conventional imaging guidance [24]. Contrasted-enhanced PET/CT optimize procedural duration time, with a mean time for image fusion of 4.6 + 1.5 min and raise the overall ablation accuracy with complete ablation in 82% of cases [25,26].

FI in the treatment of HCC has a role in difficult cases defined according to literature as lesions located in a high-risk location adjacent to critical structures and less of 10 mm from them and undetectable lesions on the B-mode US, with 99.4% of technical efficacy rate [15]. Kim et al. reported a 25.3% of non-detectable hepatic lesions on pre-treatment planning US, that, with FI, have been treated by TA.

Huang et al. compared the clinical application of CEUS, an CT/MRI fusion with CEUS and three-dimensional ultrasound CEUS. All three techniques are feasible, although FI is more appropriate for multiple tumors, risky tumors and ALBI grade 2 and 3 [6].

FI reduces the major complications during liver ablations giving an optimal view of the lesion and the surrounding structures.

### 4.2. Kidney

TA is a proven treatment for small renal lesions; it offers the advantage of limiting the invasiveness and the mortality and of preserving the renal function [27]. Mainly, TA is useful in patients with a centrally located tumor, where surgery would imply total nephrectomy [28,29]. The widely used imaging modalities for percutaneous ablation are CT and US. FI seems the optimal guidance for his capability to merge the advantages of both techniques and overcome the weaknesses of every modality [30,31]. The advantages of FI are identifying the isoechoic lesions and reducing major complications due to a better field of view [32,33].

US guiding for TA is widely used for real-time capability and the broad availability in interventional suites. While CT is crucial to precisely evaluate the anatomical relationship between the lesion and surrounding organs. TA ideal setting is hybrid US-CT/angiography suites [34].

FI is used to co-register previous CT or CBCT and real-time US to perform tumor ablations. Assisted CBCT and US FI are effective, safe and feasible in TA, but only a few papers investigated the advantages and limitations of these techniques [35,36,37]. A case of a patient with renal tumor treated with the guidance of FI is shown in Figure 3.

### 4.3. Lung

For diagnosis in many pulmonary diseases, percutaneous image-guided needle biopsy has proven to be a safe and effective technique gaining a pivotal role in the diagnostic workflow [38]. In addition, TA is increasingly used in the treatment of cancer patients, particularly in slow progressing oligometastatic patients, where radiation therapy is not feasible. CT is the most used imaging for lung biopsies and TA, but it does not allow the real-time visualization of the needle. This issue can be overcome using CT fluoroscopy which is currently more and more used to guide percutaneous biopsies and ablations of pulmonary nodules but is associated with significant radiation exposure both to the patient and the operator and it is not available in all radiology services [39].

Another technique that has had an increasing role is CBCT that produces CT images with the possibility of real-time fluoroscopy visualization. It offers greater flexibility in the orientation of the detector around the patient compared to the closed CT gantry [40].

FI technique is recently being applied to lung procedures since software algorithms were developed to register the intraprocedural anatomic imaging with preprocedural PET/CT data. Some studies reported the experience of fusion PET/CT-CBCT studying it in guiding percutaneous biopsy of lung lesions compared with CBCT guidance [40,41,42]. PET/CT-CBCT-guided is feasible and safe as CBCT-guided biopsy with a lower number of non-diagnostic samples and false-negative cases, with evidence of metabolic information provided by a prior PET/CT and a better quality of the samples obtained with FI guidance [40]. Systems that allow to guide a virtual needle on a pre-acquired lung CT have been implemented to reduce the number of CT acquisition required in lung procedures and to improve the non-axial approach in CT guided lung ablations. A case of a lung nodule treated with microwave ablation is shown in Figure 4.

### 4.4. Musculoskeletal

FI has been recently applied also in the musculoskeletal field to guide biopsies and ablations.

FI difficulties should theoretically be minimal in musculoskeletal procedures since the bone is not a mobile structure compared to other target organs. However, because of differences in patient positioning and potential patient’s movements, the operator must look for any significant deviation during the procedure in usual practice [43].

FI has been used to perform joint and perineural injections as described by Sartoris et al. which demonstrated that US needle guidance with MRI fusion assistance allows for safe and effective injection of degenerative facet joint disease [44]. Furthermore, FI of real-time US and CT scans is feasible to guide needle insertion into the Sacro-iliac joint [45].

In fact, several papers have focused on bone or soft-tissue biopsies [46]. US fusion technique with previously obtained CT or MRI data provides a high diagnostic yield and accuracy comparable to CT-guided biopsy. FI has the advantages of faster scheduling and biopsies time and the procedure is more comfortable both for the operator and patient [46].

Garnon et. Al. demonstrated the technical feasibility and safety of combined FI and needle tracking under US guidance to target bone lesions without cortical disruption [43]. The authors highlight the mandatory use of a tracking device to confirm the trocar tip’s exact location [43].

FI has also been tested in spine procedures. CT/US FI was demonstrated to be feasible and safe in guided bone biopsy of spinal lesions; this technique allows for continuous needle monitoring and image acquisition, avoiding the need for repeated CT scans and decreasing the number of CT passes throughout the procedure with shorter lead times compared to standard CT guidance [47]. A case of a bone ablation performed with the assistance of FI is shown in Figure 5.

### 4.5. Prostate 

Transrectal US-guided biopsy technique (TRUSbx) was introduced nearly 30 years ago and it has been the standard-of-care to either confirm the diagnosis or exclude the presence of disease [48]. The major disadvantages of the TRUSbx are the systematic sampling errors [49]. These limitations have highlighted the need to improve the diagnostic information gained by the invasive prostate biopsies and maximize the detection of significant prostate cancers (sPCa), while minimizing the over-detection of insignificant diseases [50]. To overcome this issue, growing evidence support the use of multiparametric MRI (mpMRI) to perform targeted biopsies (TBx) [51]. In fact, the gold standard of MRI-guided TBx is the in-bore TBx in the MRI suite which can accurately sample lesions of interest with direct image confirmation of needle deployment within the target [52]. This technique is associated with not negligible costs, limited availability and does not allow for concurrent systematic sampling [52]. The application of FI, combining mpMRI data with TRUS (MRI/TRUS fusion) joins mpMRI images and the more comfortable US guidance. It demonstrated to be time and cost-saving, while preserving an adequate targeting accuracy [53,54]. An additional key advantage of MRI/TRUS fusion is that TBx can be combined with systematic biopsies, as recommended by the European Association of Urology (EAU) guidelines [55]. Several commercial software platforms have been developed to increase targeting accuracy that differ in both, technology (image acquisition and tracking mechanism) and biopsy route (transrectal or transperineal) [50]. TA is not routinely applied at prostate tissue, but experiences are increasing in this field, which represents a very promising field of application of TA for the future [56,57]. A case of biopsies of a prostatic nodule visible on MRI is shown in Figure 6.

## 5. Limitations

There are some limitations of FI techniques that should be taken into consideration. One of the most critical problems is the potential mistargeting of US-invisible lesions. Few works deeply analyzed this limit; Lim et al. reported the 1.3% incidence of mistargeting after FI-guided radiofrequency ablation procedures of HCC [13,58]. The mistargeting in FI based on pre-acquired images is also due to the deformation of organs owing to differences in patients’ pose and variations in slice thickness of images [59].

An additional potential problem of FI is related to the respiratory motions of some target organs. In addition, some organs like kidney are often displaced by the ablative device when it is inserted into the patient body. Thus, real-time US visualization and availability of cross-sectional imaging as CT or CBCT in the operatory theatre remain of paramount importance for the success of the procedure [60]. However, these problems can be mitigated with operator expertise, software development and controlled respirations techniques like intubation or one-lung ventilation [16].

## 6. Conclusions

In many interventional oncology fields, FI improves the detectability of poorly visible lesions since it has the capability of merging multiple imaging, enabling to take advantage of each modality pros and overcoming each one cons, increasing the correct identification and targeting of lesions.

FI guidance increases the operator confidence in the ablation procedures due to better visibility of lesions and surrounding structures leading to an expansion of the indications of ablation treatment [6].

Furthermore, FI increases TA accuracy, feasibility and safety, displaying the positions of lesions and their relations with the closer organs [4].

The current literature supports the use of FI in ablation procedures to expand treatment feasibility, improve operator confidence, therapeutic effects and procedures safety.

Thus, in conclusion, we envision a scenario where FI will be more and more present in the interventional suites and largely applied in the guidance of a variety of TA in different clinical situations. Further technological advancements and studies on clinical application of FI are expected to support the diffusion of this technique in clinical applications.

## Figures and Tables

**Figure 1 diagnostics-11-00549-f001:**
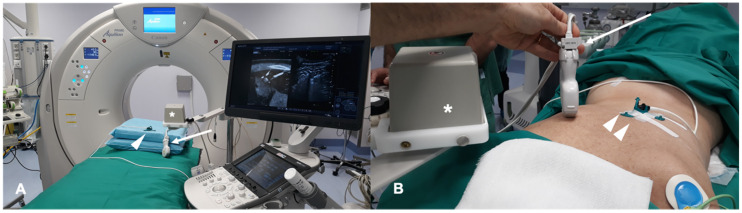
Operating room equipment for fusion imaging: (**A**,**B**) Computed Tomography (CT) scan gantry and ultrasound system for automatic image registration of fused images thanks to a disposable locating device visible under CT (white arrow heads), an ultrasound probe sensor (white arrow) and electromagnetic generator (white asterisk).

**Figure 2 diagnostics-11-00549-f002:**
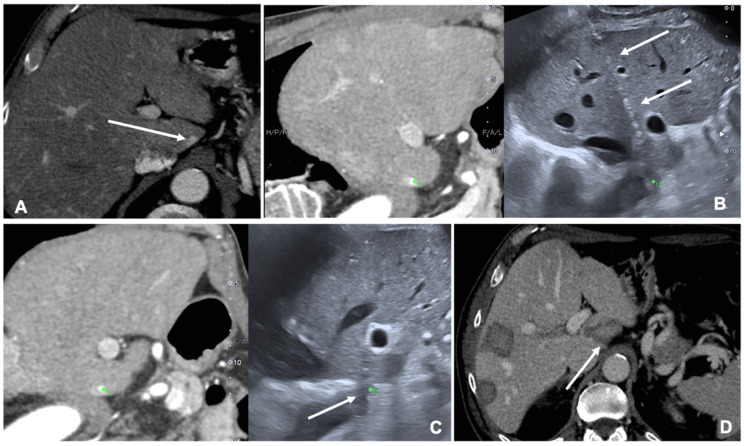
Case of a patient with liver metastases from renal cancer treated with microwave ablation (MWA). (**A**) Axial view during portal phase of the preoperative contrast enhanced Computed Tomography (CECT) showing a 6 mm enhancing lesion of segment I (white arrow); (**B**) Live multiplanar reconstruction of the same source CT (**left** side) using fusion imaging (FI) with intraoperative ultrasound (**right** side). In both views target lesion is marked by a green spot while the MWA needle is marked by two solid arrows in the ultrasound image (**right**). (**C**) FI during MWA of the lesion in the liver segment I. Arrow points out the area of gas formation at the tip of the ablation needle. (**D**) An axial view of the CECT control the day after the procedure. Arrow points out the area of ablation of the liver segment I.

**Figure 3 diagnostics-11-00549-f003:**
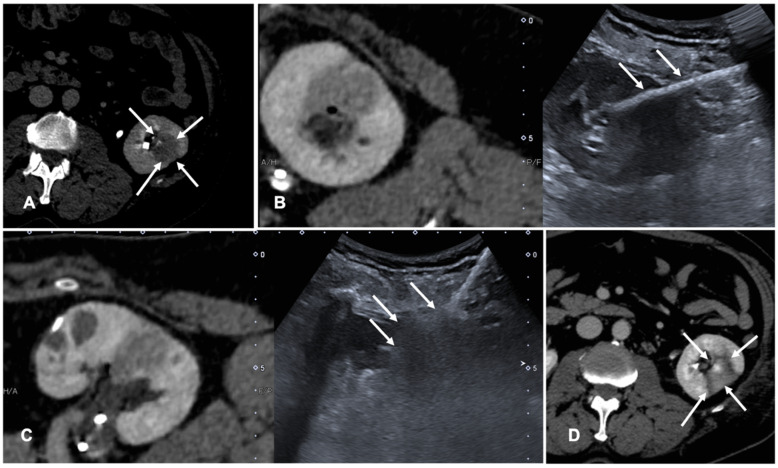
Case of a patient with left kidney clear cell renal cancer treated with microwave ablation (MWA). (**A**) Axial view during delayed excretory phase of the preoperative contrast enhanced Computed Tomography (CECT) showing a 25 mm intrarenal lesion (white arrows); (**B**) Fusion imaging (FI) of the same CT (**left** side) and intraoperative ultrasound (**right** side). MWA needle is marked by two solid arrows on the right. (**C**) FI during MWA of the lesion. Arrows point out the area of gas formation at the tip of the ablation needle. (**D**) An axial view of the control CT at the end of the procedure. Arrows point out the resulting completely ablated area of the kidney lesion.

**Figure 4 diagnostics-11-00549-f004:**
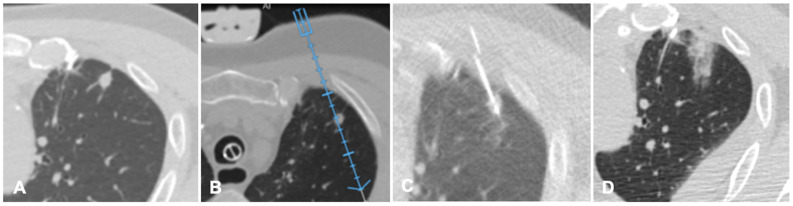
Case of an oligometastatic colonic adenocarcinoma in a patient with a nodule in the upper lobe of the left lung treated with microwave ablation (MWA). (**A**) Axial view of the preoperative Computed Tomography (CT) scan confirms the presence of a 4 mm nodule in the left upper lobe adjacent to the pleural profile. (**B**) Using electromagnetic virtual navigation guidance, the skin entry point is selected and the MWA antenna is inserted. (**C**) Multiplanar reconstruction of intraoperative CT scan along the needle long axis showing correct targeting of the lesion. (**D**) Control CT at the end of MWA treatment showing complete ablation without complications.

**Figure 5 diagnostics-11-00549-f005:**
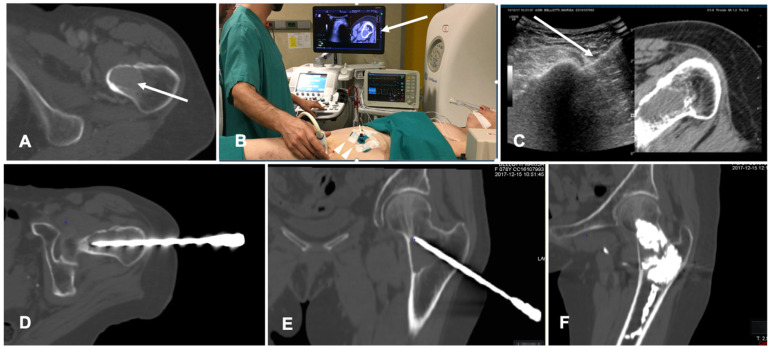
Case of a patient with a breast cancer femoral neck metastasis treated with cryoablation. (**A**) Axial view of the preoperative unenhanced Computed Tomography (CT) showing a lytic ovoid femoral neck lesion (white arrow); (**B**) Intraoperative setting for fusion imaging (FI): ultrasound (US) system with automatic image registration of fused images with previously acquired CT volume data sets (white arrow), a disposable locating device and sensor (white arrowheads) and electromagnetic generator (white asterisk). (**C**) Live FI with CT multiplanar reconstruction (right side) and US (left side). With combined CT-US imaging the needle (white arrow) can be oriented along the major axis of the bone lesion regardless the acoustic barrier. (**D**,**E**) A paraxial view and a coronal view along the needle length of the intraoperative CT confirming correct positioning into the lesion. (**F**) Final control CT on coronal plane after cryoablation and cementoplasty.

**Figure 6 diagnostics-11-00549-f006:**
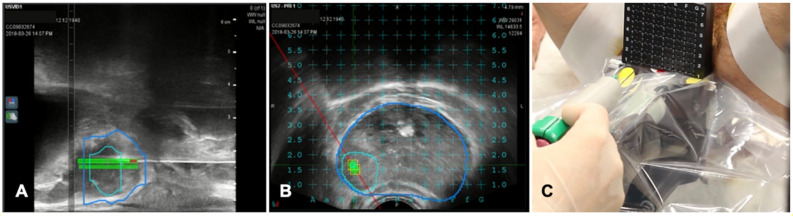
Case of biopsies of a magnetic resonance imaging (MRI) visible prostate nodule. (**A**) Parasagittal long axis view of intraprocedural ultrasound fused with regions of interest from MRI. Target lesion has been segmented in MRI images and is encompassed by the green line in the prostate volume (blue line). (**B**) Axial view of the target lesion (green line) in prostate parenchyma (blue line) derived from MRI. A reference grid is superimposed to select the appropriate space for needle placement. (**C**) A transperineal biopsy needle is then placed in the selected hole in the reference grid and advanced into the target lesion.

## Data Availability

No new data were created or analysed in this study. Data sharing is not applicable to this article.

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
