# Peer review of "Role of Fusion Imaging in Image-Guided Thermal Ablations"

_diagnostics, 2021, doi:10.3390/diagnostics11030549_

Round 1
Reviewer 1 Report
The manuscript gives an overview of the advantages of using image fusion for the treatment of different kinds of cancer. For a review paper, there is an unbalance between publications by the authors and others. This unbalance should be improved.
The origin of the images must be provided. Are they from an own study specifically for this publication? In that case, which hospital. If the images are from another publication, please cite them.
Otherwise, the manuscript reads well and is suited for publication in Diagnostics Journal.
Author Response
Reviewer 1
1. The manuscript gives an overview of the advantages of using image fusion for the treatment of different kinds of cancer. For a review paper, there is an unbalance between publications by the authors and others. This unbalance should be improved.
Author’s reply: Thank you for your suggestion, we have added the reference (ref. 2, 3, 5, 12, 17, 19, 20, 26, 29, 31, 33, 37) and improved the unbalance.
2. The origin of the images must be provided. Are they from an own study specifically for this publication? In that case, which hospital. If the images are from another publication, please cite them.
Author’s reply: All images are from authors’ institution (European Institute of Oncology), and all patients provided informed consent to the use of their images for publication.
Reviewer 2 Report
The work is interesting and well written. In my opinion you could be add some images about prostate and lung FI.
Author Response
Reviewer 2
1. “ you could be add some images about prostate and lung FI.”
Author’s reply: Thank you for pointing this out, we have add the images about prostate and lung FI.
Reviewer 3 Report
This is a very interesting and exhaustive review on a key topic in oncology and interventional radiology. I have no criticisms about the manuscript and its draft.
Just a minor comment:
P1, LINE 39 :Rephrase to ionizing radiation
Author Response
Reviewer 3
1. P1, LINE 39 : Rephrase to ionizing radiation
Author’s reply: Thank you for the comment we rephrased it.